# Breathing Control Exercises Delivered in a Group Setting for Patients with Chronic Obstructive Pulmonary Disease: A Randomized Controlled Trial

**DOI:** 10.3390/healthcare11060877

**Published:** 2023-03-17

**Authors:** Sibylle Cazorla, Yves Busegnies, Pierre D’Ans, Marielle Héritier, William Poncin

**Affiliations:** 1Haute École Libre de Bruxelles Ilya Prigogine (HELB), 1070 Brussels, Belgium; 2Collège d’Ergothérapie de Bruxelles (CEBXL), 1080 Brussels, Belgium; 3Institut de Recherche Expérimentale et Clinique (IREC), Pôle de Pneumologie, ORL et Dermatologie, Université Catholique de Louvain, 1200 Brussels, Belgium

**Keywords:** occupational therapy, chronic obstructive pulmonary disease, quality of life, breathing control exercises, group exercises, energy conservation techniques

## Abstract

Breathing control exercises are an important component of occupational therapy in patients with chronic obstructive pulmonary disease (COPD). Delivering these exercises in group settings may enhance their benefits. Therefore, this study assessed the effectiveness of breathing control exercises delivered in a group format to patients with severe COPD remitting from an acute pulmonary exacerbation. This randomized controlled trial of 6 weeks’ duration compared the addition of breathing exercise sessions delivered in a group setting to a standard exercise inpatient rehabilitation program (usual care) versus usual care alone. The standard exercise program consisted of endurance and strength training and therapeutic patient education. The intervention group received, in addition to usual care, 20 sessions of 30 min duration of breathing control exercises in a group setting. The primary outcome was quality of life (Saint George’s Respiratory Questionnaire). Secondary outcomes were the COPD assessment test, modified Borg scale, handgrip strength test, and five-time sit-to-stand test. Thirty-seven patients aged 69 ± 7 years were recruited. After the 6-week period, all outcomes significantly improved and exceeded the minimal clinically important difference in the intervention group only. Between-group changes were significant for each outcome. Conclusions: breathing control exercises in a group setting provide clinically relevant benefits in patients with severe COPD who are remitting from an acute pulmonary exacerbation.

## 1. Introduction

Chronic obstructive pulmonary disease (COPD) is a common debilitating chronic lung disease characterized by irreversible airflow limitation. The most common symptoms are breathlessness, chronic cough often accompanied with phlegm, and chest tightness [1]. Dyspnea during exercise is one of the most apprehended symptoms, occurring for increasingly mild efforts as the disease progresses. To prevent exertional dyspnea and other symptoms mentioned above, many patients adopt a sedentary lifestyle which predictably leads to extensive skeletal muscle deconditioning and social isolation with its negative psychological sequalae [2]. This process results in severe limitation of exercise capacity and physical activity, which affects a person’s ability to perform activities of daily living (ADL) [3]. Accordingly, compared to healthy subjects, patients with COPD spend around half the amount of time in waking activities, but 1.4 times more of their waking hours in sedentary behavior [4,5].

The consequences of dyspnea and activity limitation severely impair the quality of life (QoL) in patients with COPD [6,7]. As long as no cure exists, improving QoL in those patients has become the main objective of currently available treatments [8]. In this respect, pulmonary rehabilitation is a multidisciplinary and comprehensive intervention for patients with COPD with undisputable evidence of effectiveness in relieving dyspnea, improving patient participation in ADL, enhancing self-efficacy to manage the disease, and improving QoL [9,10]. Therefore, given their scope of practice, occupational therapists are key players in pulmonary rehabilitation programs. Indeed, occupational therapy intends to improve the self-management of patients’ condition (autonomous use of oxygen therapy, decreased energy expenditure, awareness of situations at risk of shortness of breath) and to enhance participation in daily life by minimizing breathlessness during these activities. Different aspects of education and training in energy conservation methods are also provided to the patient [11].

Previous studies have shown that pulmonary rehabilitation programs including occupational therapy and breathing techniques were effective in reducing dyspnea and improving the performance of ADLs and QoL in patients with COPD [12,13,14]. However, to the best of our knowledge, no data exist regarding the effectiveness of breathing control exercises delivered in a group setting. It has already been demonstrated that groups quickly influence individual behavior and promote physical activity across a wide range of people [15]. For years, there has been considerable interest in exploring means to increase participation in physical activity in COPD patients [4]. Compared to individual training, exercising in a group may increase participation in physical activities and optimize the QoL gains through socialization with other people in the group [16].

Therefore, the objective of this study was to assess the effects of breathing control exercises delivered in a group setting by occupational therapists on patients with COPD.

## 2. Materials and Methods

### 2.1. Participants

Patients with COPD stage III or IV according to the Global Initiative for Chronic Obstructive Lung Disease (GOLD), hospitalized in the J. Bracops Hospital (Brussels, Belgium) and recovering from an acute pulmonary exacerbation, were scheduled to start an inpatient rehabilitation program in the dedicated center of J. Bracops Hospital. Exclusion criteria were the following: heart or orthopedic complications; active smoking; and inability to understand instructions due to language barriers or cognitive impairments. The study protocol was approved by the local ethics committee (no. CEHIS/2019-16). All included participants provided informed consent to participate in the study. The trial was registered in ClinicalTrials.gov (NCT05199987).

### 2.2. Study Design

The study was designed as a randomized controlled trial. Patients were randomly assigned to receive either the usual care (standard post-exacerbation training program as used in J. Bracops Hospital involving aerobic and strength training as well as individual occupational therapy sessions—control group), or occupational therapy sessions in a group setting focusing on breathing control exercises in addition to usual care (intervention group). Regardless of the group, the program length was 6 weeks, during which standard exercise training sessions were scheduled once a day, 7 times a week. Randomization was prepared by a clinician not involved in this trial in a 1:1 ratio. Allocation concealment was prepared by an independent researcher not involved in the trial. The study clinician was blinded to the patient’s assignment group during the baseline assessment battery. After these baseline tests, the clinician was unblinded as it was not possible to maintain blinding during treatments.

### 2.3. Control Group

The standard inpatient training program at J. Bracops Hospital consisted of endurance training, upper and lower limb strength training, individual (one-to-one) therapeutic patient education sessions including energy conservation techniques, and ADL toilets.

Endurance and strength training components were scheduled 7 days a week and always supervised by a physiotherapist. Endurance training consisted of continuous walking on a treadmill or biking on a cycle ergometer for 30 min. The intensity was individually based on the modified Borg scale score which had to be between 4 and 6 out of 10. Upper and lower limb strength training was performed exclusively on resistance machines for 30 min. Patients had to do 3 sets of 10 repetitions for each targeted muscle group. The visual analogic scale was used to set the intensity of exercise, which had to be between 4 and 6 out of 10.

Therapeutic patient education consisted of teaching the anatomy of the lungs, the pathophysiology of COPD, and the proper use of prescribed inhaled drugs. Therapeutic patient education also included walking (5 times a week, 15 min per session) or stair climbing (twice a week, 30 min per session) activities in which patients learned to cope with dyspnea by fractionating their efforts. During walking-related tasks (e.g., tying the shoelaces by putting the feet up on the chair, balancing the load in a shopping bag) [17], patients were also instructed to control their breathing pattern using pursed-lip breathing (inhaling slowly and deeply through the nose, maximizing the recruitment of the diaphragm, and exhaling through the mouth with pursed lips), and to use economical or ergonomic postures.

Finally, patients received 2 sessions of ADL toilets (20 min per session) provided by an occupational therapist during their program. The ADLs are based on gestural learning, including the repetition of a gesture, aimed at refining it to make it more economical and less constraining [18]. The first session of ADL toilets was provided during the first week of the program and included the assessment of the autonomy and independence of the patient, advice on fractionating the efforts while toileting, and proposal of technical aids for toileting activities if necessary. The second session was provided during the third week of the program and was primarily used to verify whether the advice and techniques provided during the first ADL have been incorporated. Adjustments were proposed when needed.

### 2.4. Intervention Group

In addition to the usual care described in the control group, patients randomized in the intervention group received 3–4 sessions per week (for a total of 20 sessions per program) of breathing control exercises in a group setting (5–6 participants per group), lasting 30 min per session. The latter sessions were provided by an occupational therapist. Breathing control exercises are defined as any breathing technique that can allow deeper inspiration or expiration, or otherwise alter the rate, pattern, or rhythm of respiration [19]. Breathing techniques were incorporated during 15 different sets of body movements that had to be each repeated 10 times, with a recovery time of 10 s between each body movement (Appendix A). These movements were all open-chain upper-body and lower-body kinetic exercises, except for the sit-to-stand movement. Free weights (dumbbells, weighted balls, etc.) were given to the patients with two aims: to improve the patients’ muscular condition and to slow down the requested motions, thereby increasing the awareness of the different phases of movement (concentric phase in inspiration, eccentric phase in expiration) and helping the patients to adapt their breathing to the effort. The free weight loads of each exercise gradually increased over the sessions whenever the patient was able to perform 10 repetitions comfortably. Body movements were chosen according to their similarity with activities carried out in daily life (e.g., passing one’s hand in the back to wash, catching a product at height, bending down to tie one’s laces, etc.).

### 2.5. Outcome Measures

Outcomes recorded before and at the end of the 6-week trial period were QoL, handgrip (HG) strength, functional status, and dyspnea. Quality of life was assessed via the French version of the Saint George’s Respiratory Questionnaire (SGRQ) [20] and the COPD Assessment Test (CAT). The SGRQ measures overall health, daily living, and perceived well-being across three domains: “symptoms”, “activities”, and “impact”. The score for each domain and the total score ranged from 0 to 100, with 100 indicating a very poor quality of life [21]. The CAT consists of eight questions, with a score associated with each question ranging from 0 to 5, with 0 indicating no impact on quality of life. The total score ranged from 0 to 40. The higher the score, the greater the impact on quality of life [20].

Handgrip strength was assessed via a Jamar^®^ hand dynamometer. The best value of three repetitions of the best hand was reported [22]. Regarding the functional status, the five-time sit-to-stand test (5STST) was used. The time the individual takes to sit down and stand up completely over five repetitions was reported [23]. Dyspnea intensity was assessed immediately after the 5STST using the modified Borg scale [24].

### 2.6. Statistical Analysis

The SGRQ score was the primary outcome. The sample size was calculated based on an assumed SGRQ population standard deviation of each of the two groups of 17.3 [25] and a large effect size (Cohen’s d = 1) based on our clinical experience. With a power of 80% and an α risk of 0.05, we estimated that 17 patients per group had to be recruited in this trial (PASS 14, NCSS, LLC, Kaysville, UT, USA). Considering a drop-out risk of 15%, we aimed to recruit 20 patients per group.

Data were analyzed using SPSS v27.0 (IBM software). The normality of data was verified with the Shapiro–Wilk test. All data were subsequently presented as median (interquartile range) because of their non-normal distribution. A comparison of variables between both groups was performed using the Mann–Whitney test. Proportions were compared using the chi-squared test or Fisher’s exact test when the expected number was lower than five. A *p*-value ≤ 0.05 was considered statistically significant.

## 3. Results

### 3.1. Participants

Between February and August 2022, 40 eligible participants were screened, of whom 37 agreed to participate in the study (Figure 1). All fulfilled the entire study protocol and were analyzed. Their demographic data and baseline characteristics are outlined in Table 1. All outcomes did not significantly differ between both groups at baseline except for the prevalence of osteoporosis.

Continuous variables are reported as median (interquartile range) while categorical variables are presented as numbers (%).

### 3.2. Primary Outcome

The SGRQ score significantly decreased (improved) in the intervention group (from a median of 60.4 [43.6–71.8] to 43.2 [35.0–56.7], *p* < 0.001) but not in the control group. The between-group difference was statistically significant (*p* < 0.001, Figure 2). The pooled mean difference of 14.4 in the intervention group exceeded the minimal clinically important difference (MCID) which is 4 points for the SGRQ [26]. Of 19 patients in the intervention group, 15 (79%) exceeded the MCID of the SGRQ score.

### 3.3. Secondary Outcomes

The CAT score significantly decreased (improved) in the intervention group (from a median of 25.0 [14.0–27.5] to 15.0 [2.0–21.0], *p* < 0.001) but not in the control group. The between-group difference was statistically significant (*p* < 0.001, Figure 2). The pooled mean difference of 8.7 in the intervention group exceeded the MCID which is estimated between 1.0 to 3.8 points [27]. Eighteen out of 19 patients randomized in the intervention group (95%) exceeded the MCID of the CAT score.

The 5STST time significantly decreased (improved) in the intervention group (from a median of 15.4 [12.0–19.0] to 12 [10.7–14.9], *p* < 0.001) but not in the control group. The between-group difference was statistically significant (*p* < 0.001, Figure 2). The pooled mean difference of 3.8 in the intervention group exceeded the MCID which is 1.7 s for the 5STST [28]. Twelve (63%) patients randomized in the intervention group exceeded the MCID described for the 5STST score.

The Borg score after the 5STST significantly decreased in the intervention group (from median 8 [7–8] to 5 [3–6], *p* < 0.001) but not in the control group. The between-group difference was statistically significant (*p* < 0.001, Figure 2). The pooled mean difference of 2.9 in the intervention group exceeded the MCID which is between 1.0 and 2.0 units for the Borg score [29]. Eleven (58%) of the patients randomized in the intervention group exceeded the MCID of the Borg score.

The HG strength significantly increased in the intervention group (from a median of 23.5 [20.5–26.8] to 25 [23.5–30], *p* < 0.001) but not in the control group. The between-group difference was statistically significant (*p* < 0.001, Figure 2). The pooled mean difference of 2.8 (unit) in the intervention group exceeded the MCID which is 1.4 kg for the HG strength [30]. Seventeen (90%) patients in the intervention group exceeded the MCID described for the HG strength score.

## 4. Discussion

This study is the first to assess the effectiveness of breathing exercises in group settings in patients with COPD. Compared to only standard care, which involved aerobic and resistance training as well as individual therapeutic patient education, patients who additionally received breathing exercises in a group setting obtained clear benefits in terms of QoL, functional capacity, or upper muscle strength. These results suggest that breathing control exercises delivered in a group setting are a reliable and credible care service for patients with severe COPD.

One of the most prominent roles of occupational therapy is to relieve individuals’ participation restrictions in performing their day-to-day tasks. During inpatient rehabilitation, therapeutic patient education is provided on an ongoing basis to promote sustainable changes in patient behavior, with the aim of maximizing autonomy and independence at hospital discharge. Therapeutic patient education includes teaching energy conservation techniques during personalized gestural exercises, taking into consideration the patient’s own environment (living environment, aids, organization, etc.) and lifestyle habits (e.g., does the patient wear Velcro or lace-up shoes?) [31]. However, patients with COPD may face significant barriers in their process of behavior change due to knowledge and skill deficits, which will hinder lifestyle modification [32]. To promote a change in behavior, integrating the demands of the disease into the patient’s daily routine is essential [32]. Proposing occupational therapy in group settings rather than via traditional individual sessions takes on its full meaning here, both in terms of improving individual efficiency and in terms of solidarity and exchanges within the group leading to social efficiency. Moreover, belonging to an exercise group facilitates engagement in physical activity and helps to reduce loneliness [33]. It has also been shown that group cohesion leads to increased adherence to an exercise program over time [34]. Therefore, the establishment of groups per se may have played an important role in explaining the large improvement in the SGRQ and CAT score observed in patients from the intervention arm. This assumption is consistent with the findings of other studies on adults with osteoporosis or with mild cognitive impairments where group exercises were associated with improved QoL or reduced depression compared to control groups [35,36].

This study also demonstrated that patients randomized in the intervention group also significantly improved their functional capacity and handgrip strength. Skeletal muscle deficiency and deconditioning in patients with COPD are well-established in the literature [37]. To restore or improve muscle mass and function, specificity, dose, and progression are key components to consider during resistance training [28,38]. In the intervention group, the resistance training modalities proposed during breathing control exercises in a group setting were consistent with the patients’ own goals and engaged several major muscle groups at the same time. In addition, because the intervention group received additional training sessions compared to the control group, the exercise dose was greater in the former group. [38,39]. Finally, the free weight loads gradually increased in the intervention group only. All these factors may explain the large between-group differences at the end of the program in 5STST and the handgrip strength. It should however be noted that breathing exercises may also improve functional capacity, as was concluded by the meta-analysis of Holland et al. [40].

The decrease in dyspnea perception in the intervention group may be explained by both the gain in muscle strength and the enhanced control of breathing patterns during exercises. On the one hand, exercise training may have improved tolerance to physical exertion, lowering the ventilatory demand for a given effort intensity, which is of particular relevance in patients with respiratory diseases [41]. On the other hand, by altering respiratory muscle recruitment and promoting beneficial changes in breathing patterns and operating lung volumes, breathing control exercises likely improved the intensity of breathlessness during effort [42]. Accordingly, Prieur et al. have demonstrated that energy conservation strategies allowed patients with COPD to complete a stair-climbing task with reduced end-of-task dyspnea, yet without affecting the total task time [43].

Regardless of the stage of the disease, exercise training has demonstrated its effectiveness in a number of outcomes in patients with COPD, such as improved exercise tolerance, muscle strength, quality of life, and reduced dyspnea and fatigue [42]. However, despite exercise training sessions being provided to patients randomized in the control group, those patients did not respond to the program. Our findings suggest that we may have compared breathing control exercises in a group setting to an ineffective exercise training program. As per the study design, the control group received the standard inpatient post-exacerbation training program used in J. Bracops Hospital. No deviation from this program was allowed. However, many components of this program did not respect the general principles of exercise prescription susceptible to elicit functional improvement over time. According to international guidelines, the prescription of exercise for patients with COPD should be individualized using the results of the person’s pre-training assessment. Foundational exercise principles such as specificity, FITT (frequency, intensity, time, type), or progressive overload, are factors of utmost importance to consider [44]. In this study, maximal strength for resistance training or maximum work rate for endurance training were not measured at baseline; load progression was not envisaged; and exercise types did not vary over time. Therefore, the program was very likely unindividualized, with an inadequate training stimulus needed to elicit appropriate muscular adaptation.

We acknowledge several limitations in this study. First, as the control group did not receive a proper pulmonary rehabilitation program, we are not able to determine the real effect of adding breathing exercises in a group setting to a well-rounded exercise training program. Furthermore, this lack of adequate muscular stimulation in the standard care program may have amplified the true effect of our intervention group. For instance, despite inspiratory muscle training being considered effective in isolation [45], the addition of inspiratory muscle training to a well-defined and effective pulmonary rehabilitation program did not provide additional benefits in patients with COPD [46]. Second, the design of this study does not allow us to determine to what extent the observed benefits are attributable to the effect of group training or breathing exercises, or how these components mutually interact. Future studies should address these questions. Finally, the study clinician was not blinded for the final assessments, which is a clear limitation of the present study.

## 5. Conclusions

This randomized controlled trial in patients with severe COPD attending an inpatient post-exacerbation exercise program demonstrated that breathing control exercises, provided in a group setting, lead to clinically relevant physiological and psychological changes in COPD. Future studies should validate our findings in other settings (e.g., in outpatient settings or in patients with less severe COPD) and investigate key long-term outcomes such as the frequency of exacerbations, hospital admission rate, or mortality.

## Figures and Tables

**Figure 1 healthcare-11-00877-f001:**
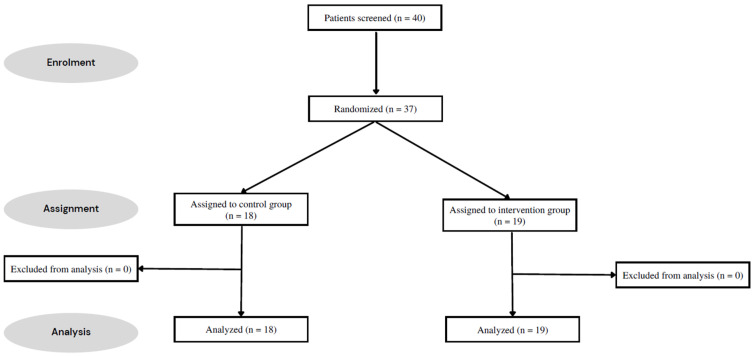
Consort flow chart.

**Figure 2 healthcare-11-00877-f002:**
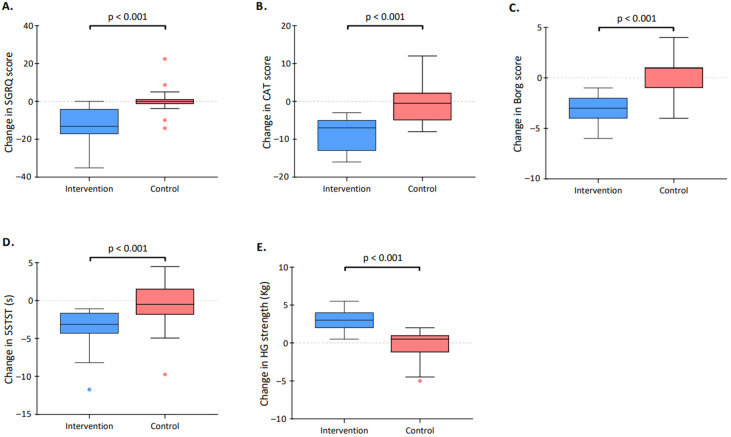
Change in clinical parameters over the study period. The figure describes the pre–post difference in both the control group (red) and the intervention group (blue) for the (**A**) Saint George’s Respiratory Questionnaire (SGRQ) score; (**B**) COPD assessment test (CAT) score; (**C**) Borg score; (**D**) five repetitions of the sit-to-stand test (5STST); and (**E**) handgrip (HG) strength. The boxes indicate the 25th and 75th percentiles; horizontal lines within boxes indicate the median; whiskers indicate the highest and lowest values within the 1.5× interquartile range; and dots beyond the whiskers indicate outliers.

**Table 1 healthcare-11-00877-t001:** Demographics and clinical parameters at baseline.

	Intervention Group	Control Group	
	n = 19	n = 18	*p* Value
Demographic data			
Age, years	67 [63–71]	68 [63–75]	0.760
Female sex, n (%)	8 (42)	7 (39)	0.842
BMI, kg/m^2^	20.9 [20.2–22.4]	22.6 [18.9–27.5]	0.261
Comorbidities, n (%)			
Gastroesophageal reflux	7 (37)	4 (22)	0.476
Chronic heart failure	8 (42)	8 (44)	0.886
Angina/myocardial infarction	6 (32)	6 (33)	0.909
Diabetes	6 (32)	3 (17)	0.447
Lung malignancy	2 (11)	0 (0)	0.486
Osteoporosis	2 (11)	11 (61)	0.002
Hypertension	10 (53)	7 (39)	0.402
Lung function			
FEV_1_ (% pred)	33.0 [26.0–44.5]	33.5 [24.0–48.0]	0.951
FEV_1_/FVC (%)	56.0 [42.5–60.0]	49.0 [37.0–60.0]	0.484
RV (% pred)	160.0 [155.0–206.5]	171.0 [157.0–205.0]	0.637
Clinical characteristics			
Dyspnea (Borg 0–10)	8 [7–8]	8 [8–8]	0.349
5STST (s)	15.4 [12.0–19.0]	15.1 [13.5–26.4]	0.412
HG strength (kg)	23.5 [20.5–26.8]	23.3 [17.5–27.5]	0.715
CAT score	25.0 [14.0–27.5]	22.5 [18.0–26.0]	0.808
SGRQ score	60.4 [43.6–71.8]	61.2 [47.6–72.0]	0.927

Abbreviations: BMI, body mass index; FEV_1_, forced expiratory volume in the first second; FVC, forced vital capacity; RV, residual volume; 5STST, 5-time sit-to-stand test; HG, handgrip; CAT, COPD assessment test; SGRQ, Saint George’s Respiratory Questionnaire.

## Data Availability

The data that support the findings of this study are available from the corresponding author, S.C., upon reasonable request. Informed consent was obtained from all subjects involved in the study.

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
