# Peer review of "Breathing Control Exercises Delivered in a Group Setting for Patients with Chronic Obstructive Pulmonary Disease: A Randomized Controlled Trial"

_healthcare, 2023, doi:10.3390/healthcare11060877_

Round 1

Reviewer 1 Report

The methodology is clearly set, the patients in the control and intervention groups are well compared.

I think that the main limitation of the study is that the patients in the control group had a different way of physical training, which reduces the importance of the conclusion related to the breathing technique in improving the outcomes that were monitored.

Reviewer 2 Report

Dear Editor,

I would like to thank you for placing your trust in me by extending the invitation to undertake this review. I find this an interesting and informative manuscript and I would recommend that it be published. However, I think there are some aspects that could be clarified:

-Has there been any loss during the follow -up?

-Why in Figure 1 (Consort Flow Chart) appears duplicated in each arm (quadruple) that is excluded from the analysis 0 people?

-Is SGRO a validated test in the language of the study?

-Why has SGHO chosen to evaluate the QoL? Is it really the most indicated test to evaluate it?

Discussion line 251-If other studios have shown that group cohesion leads to increase adherence to an exercise progression over time, why was not the control group groupal?

Minor:

-line 58: please check the verb: was-were

-The acronym SGRO should be detailed in its meaning the first time the term appears in the manuscript

Round 2

Reviewer 2 Report

In my opinion, it is an interesting manuscript that should be published